# A Century of Fractionated Radiotherapy: How Mathematical Oncology Can Break the Rules

**DOI:** 10.3390/ijms23031316

**Published:** 2022-01-24

**Authors:** Nima Ghaderi, Joseph Jung, Sarah C. Brüningk, Ajay Subramanian, Lauren Nassour, Jeffrey Peacock

**Affiliations:** 1Department of Biomedical Engineering, University of Minnesota Twin Cities, Minneapolis, MN 55455, USA; ghade001@umn.edu (N.G.); jungx567@umn.edu (J.J.); 2Machine Learning & Computational Biology Lab, Department of Biosystems Science and Engineering, ETH Zurich, 4058 Basel, Switzerland; sarah.brueningk@bsse.ethz.ch; 3Swiss Institute for Bioinformatics (SIB), 1015 Lausanne, Switzerland; 4Department of Radiation Oncology, Stanford University, Stanford, CA 94305, USA; ajayss@stanford.edu; 5Department of Radiation Oncology, University of Alabama Birmingham, Birmingham, AL 35205, USA; nassour@uab.edu

**Keywords:** fractionated radiotherapy, mathematical oncology, evolution, radioresistance, altered fractionation, intratumor heterogeneity

## Abstract

Radiotherapy is involved in 50% of all cancer treatments and 40% of cancer cures. Most of these treatments are delivered in fractions of equal doses of radiation (Fractional Equivalent Dosing (FED)) in days to weeks. This treatment paradigm has remained unchanged in the past century and does not account for the development of radioresistance during treatment. Even if under-optimized, deviating from a century of successful therapy delivered in FED can be difficult. One way of exploring the infinite space of fraction size and scheduling to identify optimal fractionation schedules is through mathematical oncology simulations that allow for in silico evaluation. This review article explores the evidence that current fractionation promotes the development of radioresistance, summarizes mathematical solutions to account for radioresistance, both in the curative and non-curative setting, and reviews current clinical data investigating non-FED fractionated radiotherapy.

## 1. Introduction

Radiotherapy is involved in 50% of all cancer treatments and 40% of cancer cures today [1]. Most of radiation is delivered in treatment fractions over days to weeks. Fractionated radiotherapy became established in the 1920s, pioneered by Henri Coutard. Hayes Martin summarized this treatment paradigm in his 1935 paper, “The Fractional or Divided Dose Method of External Irradiation in the Treatment of Cancer of the Pharynx, Tonsil, Larynx and Paranasal Sinuses” [2]:
“(1) The treatments should be given daily (or at least at short intervals), and should be of equal quantity, unless the clinical course indicates a raising or lowering of the daily doses. (2) A total treatment period of a definite length (15–20–30 days, etc.) should be decided upon, in which to deliver a certain total dosage. This treatment period should be adhered to, unless the clinical course indicates that it should be shortened or lengthened.”

Modern fractionated radiotherapy still adheres to these principles (defined hereafter as Rule One (fraction size) and Rule Two (total treatment period), respectively) established over a century ago despite drastic changes in our understanding of radiobiology [3,4,5]. The core aim of radiotherapy fractionation is the creation of a therapeutic window by leveraging differences in radiobiological principles between tumor and normal tissue. These principles can be summarized by the “5Rs of Radiobiology”, namely, Radiosensitivity, Repair, Reoxygenation, Redistribution, and Repopulation [6,7]. As an example, fractionation decreases both acute and late toxicity of normal tissue by utilizing normal tissue’s superior DNA repair capacity between fractions [8,9]. Fractionation also promotes reoxygenation and cell cycle redistribution between fractions to increase tumor radiosensitivity [10,11]. However, by prolonging the overall treatment time, it also allows for repopulation between fractions [12,13]. An optimal fractionation schedule hence balances the impact of tumor re-sensitization, regrowth, resistance onset, and advantages of normal tissue repair [14].

Current fractionated radiotherapy’s best approach is to deliver daily fractions of equal dose (Rule One), or fractional equivalent dosing (FED). FED is optimal only if tumor radiosensitivity remains constant during treatment. Factors such as intratumor heterogeneity and natural selection likely select for tumor cells undergoing FED that are more radioresistant to the dose delivered [15,16]. This can occur through the selection of de novo resistant populations and/or through acquired resistance [16,17,18,19]. In addition to FED, current radiotherapy is delivered in a short-predefined time period (Rule Two) at maximum tolerable dose aiming for tumor eradication. This is suboptimal in the setting of incurable disease where options for re-treatment are limited [20].

One reason radiotherapy may maintain century-old dogma is the infinite permutations of dose per fraction and fractionation intervals that could comprise a radiation treatment schedule, making it difficult to identify an optimal starting point. Where does one start? How do we deviate from fractionation dogma that has stood the test of time? The field of mathematical oncology provides an excellent platform to tackle these questions through in silico analysis [21,22,23,24,25]. Mathematical frameworks can efficiently mine and optimize this parameter space, and, hence, could pave the way towards clinical testing of the most promising approaches.

This review will discuss a brief introduction to mathematical oncology and its application to optimize fractionated radiotherapy, current evidence that fractionated radiation therapy can lead to radioresistance, and the use of mathematical modeling to suggest regimens to diminish the impact of radioresistance on treatment efficacy. Finally, this review will discuss current clinical data that investigates non-FED regimens. 

## 2. Review

### 2.1. Historical Mathematical Models That Determined Radiation Dose and Fractionation

Mathematical oncology allows for complex biological systems operating under the umbrella of reasonable assumptions to be distilled to equations. These equations can provide a finite space to design in silico experiments. Potential solutions that are suggested through mathematical oncology can then be tested in in vitro or in vivo systems. This in turn can be subsequently used to calibrate and infer new models.

Attempts to parameterize the effect of radiation on mammalian cells in the 1950s to the 1960s utilized mathematical oncology [26]. Of these mathematical equations, the most clinically used and validated model is the linear quadratic (LQ) model [26]. It describes the surviving fraction (SF) of clonogenic cells as a function of a single fraction treatment at radiation dose (d [Gy]):(1)$$\mathrm{SF}=e^{\left(-\alpha d-{\beta d}^{2}\right)}$$

The two parameters of this model, α [1/Gy] and β [1/Gy^2^] characterize the radiosensitivity of the irradiated cells. The α parameter is linearly related to dose, while β is quadratically related to dose. The ratio of the two parameters, α/β [Gy], is a measure of the fractionation sensitivity of the cells: cells with a lower α/β are more sensitive to fraction size. Mathematically, the α/β ratio corresponds to the dose at which cytotoxicity from the linear and quadratic components contribute equally to the surviving fraction: αd = βd^2^. Therefore, tumors with an α/β < 2 will have a dominant quadratic (as opposed to a linear) increase in tumor cytotoxicity with an increase in dose larger than conventional fractionation (1.8–2 Gy per fraction).

In the setting of fractionation, the total dose D is delivered as *n* consecutive equal fractions of doses of d. Based on the LQ model, Biological Effective Dose (BED) [Gy] facilitates direct comparison of different fractionation schemes that result in the same SF [27,28].
(2)BED=nd1+dαβ

The LQ model has found clinical application in predicting the sparing effect of fractionated radiotherapy and comparing equivalent doses of different fractionation schedules using BED [29].

A major shortcoming of this model is the assumption that the radiosensitivity parameters (α, β) remain unchanged between and within the same tumor type during radiotherapy treatment, hence neglecting both inter- and intratumor heterogeneity [15,30,31,32]. This challenge is addressed in mathematical modeling by simulating several compartments of varying α and β values within a given tumor or patient population (Figure 1) [32,33,34,35].

With varying radioresistance and repopulation patterns, different subpopulations are predominantly selected during radiotherapy, and will eventually dominate the tumor population.

### 2.2. Does Resistance Develop during Fractionated Radiotherapy?

There is growing preclinical data demonstrating that fractionated radiotherapy can create or enrich radioresistance [36,37]. A representative example is a study by van den Berg et al., where glioma cell lines were irradiated with 60 Gy in 30 fractions. Clonogenic survival was assessed throughout fractionated radiotherapy (Figure 2A) [38]. As shown in Figure 2B,C, the plateauing of the SF after ~10 fractions suggest the onset of resistance to 2 Gy per fraction. Isolated clones from this experiment had a higher clonogenic survival following radiation compared to their parental lines.

Several other studies have evolved radioresistant cell lines selected by fractionated radiotherapy in vitro and proposed a variety of different mechanisms of action related to the Rs of radiobiology.

Repair: A number of studies investigated the importance of differences in DNA repair potential within radioresistant and sensitive cell populations [8,39,40,41]. Pre-activation of pathways associated with DNA single and double strand break repair could here be linked to radioresistance. For example, Shimura et al. demonstrated increased DNA repair capacity in radioselected clones [42]. In their study, they radiated hepatocellular carcinoma cell line (HepG2) and a glioblastoma cell line (A172) at 0.5 Gy every 12 h for 82 days. The surviving cells, 82FR-31NR, were isolated and demonstrated increased clonogenic survival to 2, 5, and 10 Gy irradiation. Further investigation of the efficient DNA damage response in 82-FR-31NR cells revealed that Protein kinase B (AKT) phosphorylation and cyclin D1 were upregulated compared to non-radiated cell lines [43,44].

Radioresistance: Depending on the specific tissue, different cell types vary in their intrinsic radiosensitivity. In the context of intra tumor heterogeneity, radiosensitivity also differs within a tumor. As an example, within a tumor cancer, stem cells (CSC) and ordinary cancer cells (OCC) may exist, but CSCs are thought to be more radioresistant [45,46,47]. Mihatsch et al. investigated lung and breast cancer cell lines to explore the evolution of tumor resistance to radiotherapy and stem cell-ness [48]. Two cell lines (A549 and SK-BR-3) were subjected to 3 or 4 Gy fractions in intervals of 10–12 days for four total fractions. The remaining radioselected cells were analyzed for their clonogenic survival. The surviving fraction after 2 Gy increased in A549 cells from 0.40 to 0.53 and SK-BR-3 increased from 0.33 to 0.40. The radioselected cell lines were further analyzed for stem cell-ness based on Western blotting for putative stem cell makers. It was concluded that the presence of the cancer stem cell marker aldehyde dehydrogenase 1 (ALDH1) also indicated radioresistance. Similarly, intratumor differences in expression of coxsackie and adenovrius receptor (CAR), a regulator of cell–cell adhesion and inflammation, was shown to result in differences in radiosensitivity. Zhang et al. established radioresistant cell lines by exposing two lung cancer cell lines, H460 and A549, to 2 Gy/fraction, once a week for a total dose of 60 Gy [49]. The radioresistant clones had higher clonogenic survival compared to the parent at 2, 4, and 8 Gy. Fraction of cells positive for CAR in radioselected H460 and A549 were significantly higher than in parental H460 and A549 (*p* < 0.05). 

Redistribution: Cell cycle stage correlates with radiosensitivity due to the variation in available DNA repair mechanisms and overall amount of DNA present in the cell. Despite variations between cell types, actively dividing (M-Phase) cells are often most radiosensitive, whereas cells in synthesis (S-phase) or quiescent state are resistant [10,50,51]. McDermott et al. used fractionated radiotherapy at 2 Gy/fraction for 30 fractions to select for radioresistant clones within the human prostate cancer cell line 22Rv1 [52]. They compared cell cycle distribution, DNA double-stranded breaks, and DNA repair capacity. When compared to wild-type cells radioresistant (RR-22Rv1) cells had significantly higher surviving fractions at 2, 4, 6, 8, and 10 Gy (*p* < 0.05). RR-22Rv1 cells were also enriched in S-phase cells, found to be less susceptible to DNA damage, and more effective at DNA damage repair compared to the wild-type cells through analysis of Comet assays. This demonstrates radioresistance evolving by increasing DNA damage repair and reassortment to S-phase.

Although biological mechanisms to confer resistance vary amongst these studies, all radioresistant clones were evolved through fractionated radiotherapy and displayed increased survival after radiation compared to the parental lines. Table 1 summarizes the selected representative examples of preclinical evidence of resistance emergence during fractionated radiotherapy.

### 2.3. Breaking Rule One—Can Altered Fractionation Account for the Development of Radioresistance?

Mathematical models have explored the emergence of resistance in FED and proposed alternative fractional dosing strategies to reduce the impact of radioresistance during treatment. 

These models generally account for intratumor heterogeneity by varying radiosensitivity parameters (α, β) [22,53,54]. Resistance is either inherent or acquired through selective pressure within specific tumor subpopulations, characterized by lower α or α/β ratios leading to increased surviving fractions to conventional fractionation. Fraction sizes can be changed during radiotherapy to capitalize on the dynamically changing radioresistance of the tumor population as a whole (see Figure 3).

Heterogenous α/β subpopulations undergoing selection during FED is demonstrated in a study by Ghaderi et al. The authors implemented a discrete, agent-based model to predict surviving fractions of tumors after irradiation. The model incorporated ten independent subpopulations with unique α and β parameters in a given non-small cell lung cancer (NSCLC) tumor. Radioresistant cell populations were inherently present within the tumor, demarcated by lower α and β, replacing the initial radiosensitive population throughout a treatment employing conventional FED. A linear daily dose ramp starting at 1 Gy to a final dose of 3 Gy was tested along with dose regimens that varied temporally and in magnitude. These regimens were shown to be more effective than standard therapy (60 Gy total, given in 30 equal fractions) by as much as 1.52-fold lower tumor surviving fraction (*p* < 0.001). When tested in a validation cohort of 57 NSCLC patients utilizing a genomic estimate for α and β for each patient, their computational surrogate for tumor size “Log Cell Count” predicted a linear correlation for overall survival and local control in cox-regression analysis (*p* < 0.001, HR = 1.32 95% CI (1.13–1.52), and *p* = 0.002, HR = 1.34 95% CI (1.11–1.56), respectively), demonstrating the importance of inter- and intra-tumor heterogeneity in radiosensitivity parameters α and β for treating cancer [18].

Ramp up scheduling was also investigated by Kuznetsov et al., who implemented a partial differential equation–based model accounting for tumor cell repopulation, re-oxygenation, and redistribution of proliferative states represented as subpopulations with varying α with a fixed β (β = α/10). An optimization algorithm based on gradient descent was employed to derive an optimal fractionation schedule that maximized tumor eradication given two constraints: (1) daily dose should be beneath the maximal tolerated dose, and (2) normal tissue or Orange at Risk (OAR) exposure in the optimized schedule should not exceed that of a standard fractionated therapy (30 equal fractions of 2 Gy, over six weeks). This optimization initially selected for resistant tumor cells by eradicating more sensitive subpopulations (first stage dose < 2 Gy), followed by dose escalation (second stage dose > 4 Gy) to maximize tumor control. This dose ramp-up strategy predicted non-uniform fractionation to be at least as effective in terms of tumor control (% eradication of initial tumor volume) as standard treatment over a range of parameters α (0.07–0.21 Gy^−1^, β = α/10). For (0.09 < α < 0.13 Gy^−1^), changing daily fractions improved tumor control compared to standard treatment [55].

Alfonso et al. incorporated intratumor heterogeneity with a continuous Gaussian distribution of (α, β) in their model that predicted surviving fraction of NSCLC and prostate cancer following irradiation. The model was calibrated based on in vitro clonogenic survival data of prostate, and NSCLC cell lines. They showed that heterogeneity of (α, β) in the model was necessary to corroborate the in vitro results. During conventional fractionation, selective pressure on subpopulations of lower α/β (α/β decreases by preferentially killing subpopulations with higher α) is purported to be the cause of the emerging resistance. The study tested the following regimens of comparable BED based on the initial α/β ratio: 2 Gy × 25 fractions, 2.4 Gy × 20, 3 Gy × 15, 4.2 Gy × 10, and 7 Gy × 5. Heterogeneity of (α, β) led up to ~2 orders of magnitude reduced final tumor cell count for higher daily dose (7 Gy × 5) compared to lower daily dose (2 Gy × 25). Even though daily fraction sizes did not vary in these simulations, this study demonstrates that intratumor heterogeneity could result in the emergence of population resistance during radiotherapy [56].

In addition to tumor control, the efficacy of OAR sparing is equally important in fractionated therapy. Parsai et al. investigated varying dose to OARs during radiotherapy to allow for increased DNA repair time. This concept is referred to as temporally feathered radiotherapy (TFRT). In their study, different treatment plans were calculated that varied based on constraints on five nearby OARs—each plan was optimized to significantly spare four of the five OAR at the cost of increased exposure of the remaining OAR. Radiotherapy was then delivered over five days per week with each day of the week using a unique plan. Their simulated results demonstrated that TFRT theoretically lowers OAR toxicity as a result of a longer overall recovery time compared to conventional fractionation [57]. This mathematical model is the basis for NCT03768856, a phase I clinical trial with five participants with head and neck squamous cell carcinoma treated using TFRT.

These studies demonstrate the evolving changes in radiosensitivity that can occur with FED and possible optimizations to fractionated therapy dose to adapt to these changes.

### 2.4. Breaking Rule Two—Can Incurable Tumor Progression Be Delayed by Delivering Intermittent Radiotherapy?

Mathematical oncology has also tackled novel radiotherapy schedules to curtail the progression of tumors rather than optimal ways to eradicate them. Tumor eradication is not always possible. For instance, in glioblastoma multiforme (GBM), due to the invasive and diffuse nature of these tumors, recurrence is certain despite numerous trials investigating multiple drug regimens and radiation dose escalation [58]. In a setting of incurable disease, delivering radiotherapy intermittently with multiple days, weeks, or months between fractions may be superior compared to a maximum tolerated dose regimen. A protracted treatment’s theoretical advantages compared to standard fractionation are as follows [59,60,61]. First, the prolonged time between fractions allows for superior OAR repair allowing dose escalation at comparable normal tissue complication rates. Second, the emergence of resistance in tumor subpopulations may be delayed due to competing sensitive subpopulation repopulation. Figure 4 gives an overview of these concepts.

In GBM, radioresistant populations exist as CSC, which have a slower growth rate and compete with fast growing sensitive cell (OCC) for resources (Figure 5). Radiotherapy predominantly affects OCC subpopulation leading to an increase in resources being available to surviving CSC which can promote their growth. This can be advantageous to slow overall tumor growth, but also makes the tumor more radioresistant as a whole. Hence, there is a delicate balance between inter-fraction timing to allow for CSC repopulation, which increases overall radiosensitivity versus increasing overall tumor growth rate. In addition to compartmental variation in radiosensitivity, it is key to account for variation in repopulation potential. In the following section, we provide an overview of representative studies investigating these effects in more depth (summarized in Table 2).

Yu et al. investigated the role of evolutionary dynamics between OCC and CSC in glioblastoma with a set of ordinary differential equations. Three optimized fractionation schedules of weekly, bi-weekly, and monthly fractions (*n* = 53, 27, 13) were proposed by Monte-Carlo–based simulated annealing under the constraint of a BED = 100 Gy for OARs. Simulation results suggested improvement in time to progression for weekly, bi-weekly, and monthly super hyperfractionated schedules of 430.5, 423.9, and 413.3 days, respectively, compared to 250.3 days for conventional fractionation (2 Gy × 30 fractions). The model also predicted no difference in time to progression for other conventional fractionation schedules, such as 1.8 Gy × 33 fractions, 1.5 Gy × 40, and 5 Gy × 10 (247.6, 249.4, 234.4 days, respectively). This study provides mathematical motivation that by increasing the duration of radiotherapy, time to regression for glioblastoma can be enhanced by enriching for more stem like populations with slower growth rates in the tumor [63].

Intermittent therapy (iRT) was also examined by Brüningk et al., where hypofractionated doses were separated by multiple (four to twelve) weeks in recurrent glioblastoma. The mathematical study was calibrated using data from a phase I clinical trial (NCT02313272) for 16 patients treated with hypofractionated stereotactic radiotherapy HFSRT (HFSRT ≥ 6 Gy × 5 daily fractions) with debulking intent concurrent with bevacizumab, and the PDL1 inhibitor pembrolizumab [20]. Tumor growth curves from pre- and post-treatment MRI data were used to extract three patient specific (i.e., intertumor heterogeneity) parameters for their mathematical model: evolution of resistance towards immunotherapy, pre-treatments tumor volume, and radiotherapy response. Two iRT (iRT, ≥6 Gy × 1 every 6 weeks) and iRT plus boost (iRT + boost, ≥6 Gy × 3 in daily fractions at time of progression) were tested in silico, demonstrating that the time to progression in iRT +/− boost was at least equal to if not greater than HFSRT in 15 out of 16 cases. Therefore, iRT theoretically could increase time to progression through dose escalation by allowing more time for OAR recovery between treatments and through controlling the onset of resistance [20].

Two studies investigated simultaneous optimizations of dose and inter-fraction timing (breaking Rule One and Two) in glioblastoma. Leder et al. conducted a unique study that examined altered fractionation optimized in silico for mouse models. They examined a wide range of fractionation schedules and compared these to conventional fractionation doses (2 Gy per fraction). The initial dose response study revealed a plateau in tumor response around 10 Gy, which was subsequently set as the total test dose for different fractionation schemes. The mathematical model was a set of ordinary differential equations accounting for intratumor heterogeneity via CSC-OCC conversion. Monte-Carlo–based simulated annealing revealed a mathematically predicted schedule (“Optimum-1”) which yielded a median overall survival (OS) in mice of 50 days vs. 33 days in standard treatment (*p*-value < 0.0001). Their model further predicted survival benefit for hyperfractionated (median OS 37.5 days) and hypofractionated (median OS 36 days) regimens which did not translate to differences in survival times in vivo (*p* = 0.14, *p* = 0.06, respectively). Changing OCC and CSC conversion rate to be time dependent was necessary to rectify the in silico and in vivo OS discrepancies. Based on this implementation, their “Optimum-2” (mathematically derived regimen with changing OCC and CSC conversion rate) schedule improved OS in animals compared to standard treatment (*p*-value <0.0001). Both Optimum-1 and 2 enriched for slower growing CSC compared to standard treatment (3.55 fold, *p* = 0.03; 2.6 fold, *p* = 0.02 respectively) [62].

Treatment delivery based on “Optimum-2” was tested for safety in NCT03557372, a phase I clinical trial with 14 recurrent GBM patients [64]. The treatment delivers radiotherapy inferred from Optimum-2 to delay tumor regrowth by enriching for stem-like cells early during treatment. This results in a more radioresistant population that is compensated by an increase in the total dose of radiation, striking a balance between minimizing the total cell number and maximizing the stemlike cell fraction at the end of treatment. 

Expanding on previous work by Leder et al., Badri et al. also examined the effect of early of intermittent and varying fractionation schedules in GBM [62,65] The clinically constrained (time required for treatment feasible in an 8 am to 5 pm time frame) optimization algorithm was performed by simulated annealing. Total fraction numbers of 15 (weekends excluded) and 21 (weekends included) were tested, each allowing for a maximum of three prescribed doses per day. A vast majority of optimal fractionation schedules were predicted for hyperfractionation early on in the week and one higher fraction delivered on the last day, resulting in slower overall tumor doubling time (1000 h vs. 325 h in the standard treatment of 2 Gy daily for five days). Optimizing for inter-fraction intervals revealed maximal tumor doubling time was affected by the modeled CSC to OCC differentiation dynamics and total treatment duration. This study came to the same conclusion as Yu et al.: time to progression could be improved by enriching for more resistant stem like cells with lower turnover rates [65].

Intermittent therapy is explored in these papers as a possible solution to delay GBM recurrence rather than attempts to cure it. Moreover, these studies show that selecting for slower growing CSC may yield a longer delay to progression. This solution was further validated in vitro by Leder et al. and was the foundation for a clinical trial in GBM patients [62].

### 2.5. Current Clinical Data Utilizing Altered Fractionation

The majority of clinical radiotherapy treatments are delivered in FED, but over the past several decades there has been increasing evidence that non-conventional fractionation (1.8–2 Gy per fraction) yields a similar or better therapeutic window compared to conventional fractionation. For numerous disease types, both hyper- and hypofractionation have been evaluated in clinical trials [66,67,68,69,70]. These efforts have led to changes in the standard of care in radiotherapy treatments for prostate and breast cancer [71,72]. However, the design of these trials was based on LQ-model estimations of conventional fractionation equivalence, and, hence, were limited given our current understanding and quantifiable estimation of the required biological parameters [73,74,75]. Despite encouraging results at the patient population level for a subset of these trials, it needs to be stressed that intratumor heterogeneity and the above discussed implications on temporal variation of competing tumor populations were not accounted for in these FED hyper- and hypofractionation settings.

One extreme example of hypofractionation frequently employed in cancer in the brain is Stereotactic Radiosurgery (SRS). SRS is a minimally invasive method which delivers a highly conformal dose of high intensity (10 to 20 Gy), resulting in irradiation damage on the tumor while sparing the adjacent OAR [76,77]. Recent studies also suggest that damage on vasculature can potentially regulate tumor cell response to radiation by causing the tumor to become more resistant [78]. SRS boost after conventional fractionation is a step towards deviating from the FED-concept which has been investigated in numerous clinical settings [79,80,81]. However, it is unclear whether studies that show clinical improvement with hypofractionated boosts represent an advantage purely from dose escalation, targeting resistant surviving subclones, or a combination of both [82].

For example, a retrospective study by Quynh-Thu Le examined 45 nasopharyngeal cancers treated with a stereotactic boost of 7–15 Gy in a single fraction following 66 Gy in 2 Gy/fraction. These patients had a 3-year local control rate of 100% [83]. Given the historical baseline for 3-year local control of ~60–70% for this patient group, this approach was deemed very promising despite the small cohort size, in particular, given that 18 of the 45 tumors represented T4 disease or advanced locally invasive disease [84]. Similarly, the phase III NCT00002708 I randomized clinical control trial compared whole brain radiotherapy to whole brain radiotherapy with a radiosurgery boost in 333 patients [85]. The radiosurgical boost arm had a survival advantage for patients with single brain metastasis (median survival time 6.5 vs. 4.9 months, *p* = 0.0393). Conversely, NCT00002545 was a phase III study that did not show a benefit to a radiosurgical boost of 16–24 Gy before 60 Gy in 30 fractions compared to 60 Gy in 30 fractions alone (13.5 months vs. 13.6 months, *p* = 0.57) [86].

Hyperfractionated boosts have also been investigated with mixed results. NCT00158652, a phase III randomized clinical trial with 840 participants investigating head and neck squamous cell carcinoma, had a control arm that compared 70 Gy in 35 fractions with 2 Gy per fraction daily compared to 40 Gy in 2 Gy per fraction followed by a 1.5 Gy twice daily regimen for 30 Gy [87]. This study showed no difference in progression free survival between the two arms (HR 1·02, 95% CI 0·84–1·23; *p* = 0·88). The ASCENDE-RT was a phase III randomized trial comprising 398 patients that showed a 9-year relapse free survival benefit for low dose rate brachytherapy (ultra hyperfractionation) boost compared to dose escalation alone (83% vs. 63%, respectively (HR = 0.473; 95% CI 0.292–0.765; *p* = 0.0022) in intermediate and high-risk prostate cancer [88].

Clinical studies investigating boost delivery following or preceding conventional FED radiotherapy may serve as a stepping stone towards changes to fractionation. Current altered fractionation clinical studies still strongly build on FED, and perhaps breaking completely free from Rules One and Two will require a framework to suggest promising regimens. The authors of this manuscript propose that mathematical oncology simulations could serve as such a framework and encourage further investigation into theoretical modeling to pave the way towards clinical translation of these promising concepts.

To the best of our knowledge, there are only nineteen patients that have been treated with mathematically informed models in phase I clinical trials that truly deviate from Rule One and Rule Two. Five patients with head and neck cancer have been treated with TFRT, varying daily dose to OARs, and fourteen GBM patients have been treated with Optimum-2, a mathematically derived algorithm varying dose and fraction interval to delay time to progression. Results from these studies were not available at the time of writing this manuscript [57,62,64].

## 3. Conclusions

Fractional equivalent dosing delivered in a finite period of short intervals is an under-optimized radiotherapy delivery paradigm established a century ago. Current evidence to deviate from equal fraction size and temporally short overall treatments remains preclinical or in phase I clinical trials. Mathematical oncology may serve as a pioneering tool to change fractionation dogma and offer potential solutions; however, these hypothesis-generating studies require further validation, particularly in clinical trials.

## Figures and Tables

**Figure 1 ijms-23-01316-f001:**
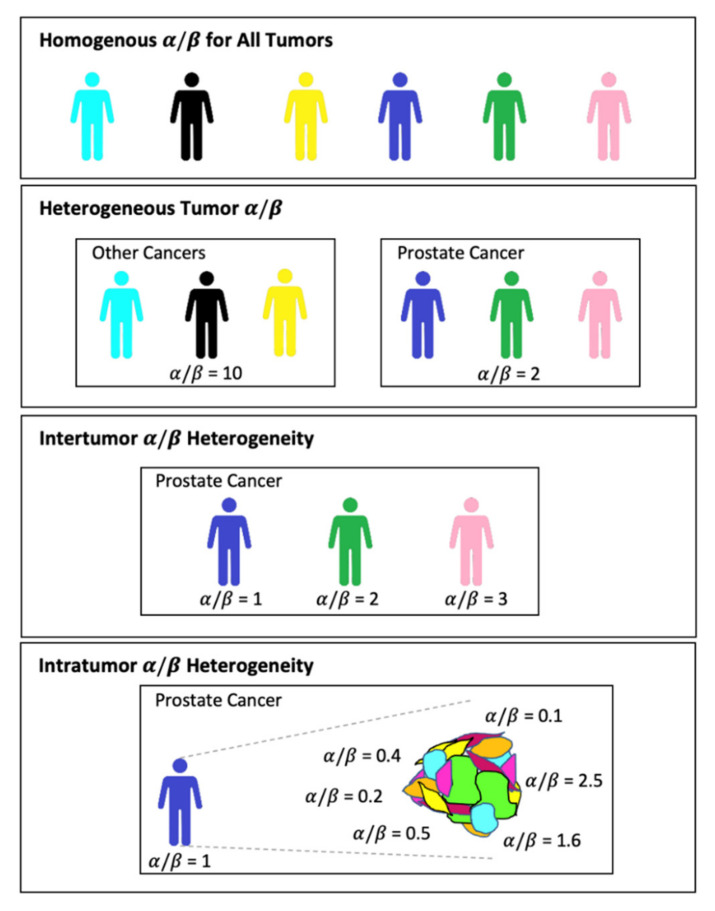
Heterogeneity of radiosensitivity parameters α/β among and within patients. Radiotherapy continues to treat under the assumption of homogeneity in α/β for most tumors with some histological specific α/β (prostate cancer as an example).

**Figure 2 ijms-23-01316-f002:**
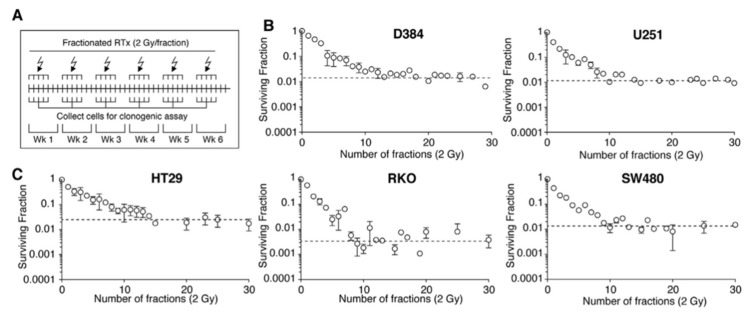
van den Berg et al. irradiated different glioma cell lines with 60 Gy in 30 fractions (2 Gy daily fraction, 5 days a week, 6 weeks) and collected cells after each fraction for clonogenic assays [38]. (**A**) A representation of the experimental set up and fractionation schedule. (**B**) Clonogenic survival for high-grade astrocytoma cells (D384, U251-MG). (**C**) Clonogenic survival for colon carcinoma cells (HT29, RKO, SW480). (**B**,**C**) After the tenth fraction, a plateau in the surviving fraction following subsequent 2 Gy/fraction was observed across all cell lines. Horizontal dotted lines represent the steady state clonogenic survival of respective cell lines after therapy. Reprinted with permission from ref. [38]. 2021 Elsevier. Abbreviation: Wk: week.

**Figure 3 ijms-23-01316-f003:**
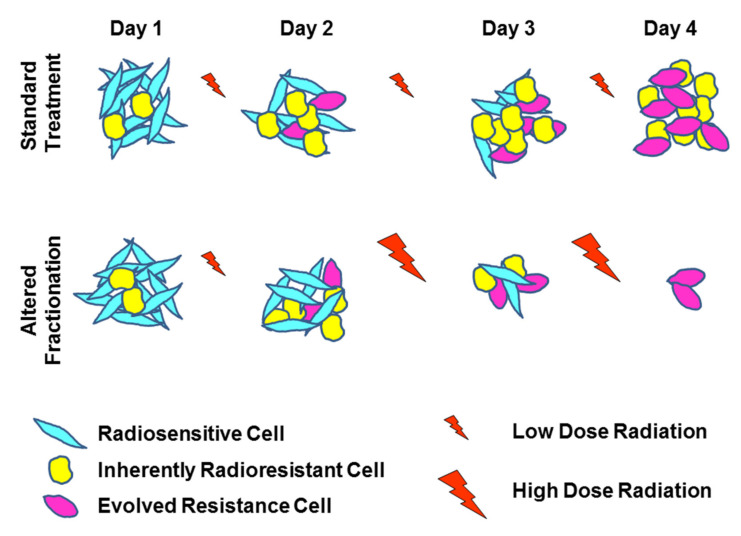
Schematic summary of altered fractionation and acquisition of resistance modeled in mathematical simulations. Using standard FED (top row) radiosensitive cells (higher α/β ratio) (teal) are preferentially killed early on, whereas radioresistant (lower α/β) subpopulations emerge (pink) or persist (yellow). Eventually, resistant phenotypes dominate the population. Increasing fractionation (ramp up schedule, bottom row) during radiation could compensate for the evolving radioresistance, leading to a higher chance for tumor eradication.

**Figure 4 ijms-23-01316-f004:**
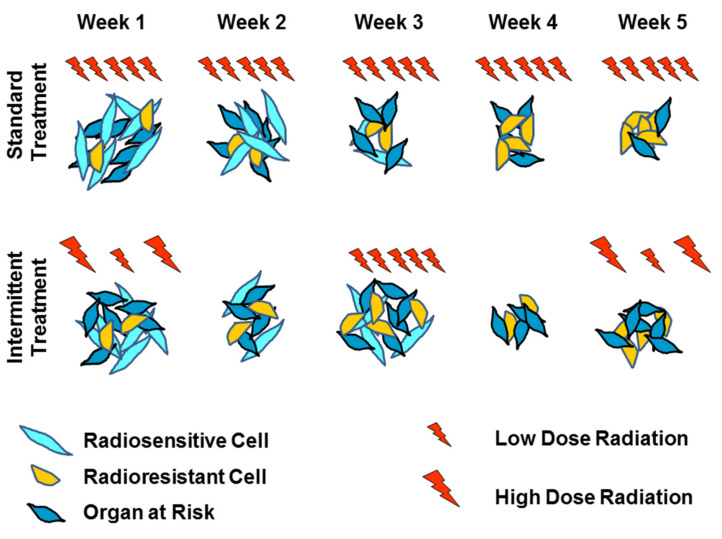
Schematic summary of optimizing inter-fraction timing. Radioresistant cells (yellow) are thought to have an increased doubling time compared to radiosensitive cells (cyan) with less DNA repair capacity of organ at risk cells (indigo). Standard FED (**top row**) given at maximum tolerance leaves a resistant population of tumor cells that will cause recurrence. By increasing the time between fractions of radiotherapy (**bottom row**), radiosensitive and organ at risk cells repopulate the environment. The top row gives radiation for curative intent while the bottom row is to limit tumor progression.

**Figure 5 ijms-23-01316-f005:**
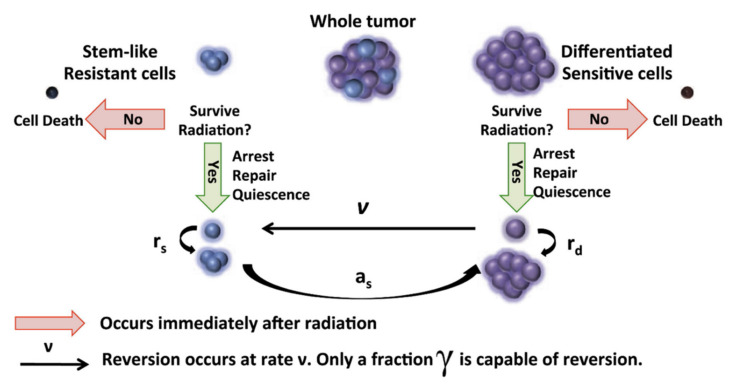
Evolutionary dynamics of PDGF-driven glioblastoma and radiation therapy modeling described in [62]. Cancer Stem Cells (CSC) and Ordinary Cancer Cells (OCC) each have their own set of radiosensitivity parameters (α_CSC,_ β_CSC_) and (α_OCC,_ β_OCC_), proliferation rates (r_CSC_, r_OCC_), and conversion turnovers (ν, a_s_), respectively. Reprinted with permission from ref. [62]. 2021 Elsevier.

**Table 1 ijms-23-01316-t001:** Selection of example studies providing preclinical evidence for the onset and underlying reasons of radioresistance following fractionated treatments. The selection includes examples covering different aspects of the principles of radiobiology.

Tumor Cell Line	Method	Findings	Reference
D384 (astrocytoma) and U-251MG (astrocytoma)	60 Gy in 30 fractions, 5 fractions a week for 6 weeks	Radioresistance is a transient feature that fades in the absence of selective pressure	[38]
HepG2 (liver) and A172 (brain)	0.5 Gy every 12 h for 82 days	DNA damage response involving AKT/cyclin D1/cdk4 pathway is preactivated in radioresistant cells	[42]
A549 (lung) and SK-BR-3 (breast)	3 or 4 Gy fractions in intervals of 10–12 days for 4 total fractions followed by Western blotting for stem cell markers	The stem cell marker ALDH1 is indicative of radioresistant cells	[48]
H460 (lung) and A549 (lung)	2 Gy/fraction, once a week for a total dose of 60 Gy followed by Western blotting for stem cell markers	The cancer stem cell marker CAR has increased expression in radioresistant clones	[49]
22Rv1 (prostate)	2 Gy/fraction for 30 fractions followed by enrichment in S phase cells	Radioresistant cells are enriched in S-phase, less susceptible to DNA damage, and acquire enhanced migration potential	[52]

**Table 2 ijms-23-01316-t002:** Summary of major findings and assumptions of discussed mathematical papers.

Key Assumptions	Findings	Cancer Type	Reference
**Breaking Rule One**			
Agent-based mode using discrete (α, β) heterogeneity within tumor and across patients, altered daily fractionation	Hypofractionation improved OS vs. standard treatment, ramp-up and uniform standard treatment have similar OS	NSCLC	[18]
PDE model (O_2_ and nutrient distribution), doubling time heterogeneity, α dependent on oxygen levels, β/α = fixed, altered daily fractionation	Non-uniform therapy improves TC vs. standard treatment (100% tumor volume reduction for 0.09 < α < 0.13 1/Gy)	Histologically agnostic	[55]
Continuous Gaussian (α, β) and doubling time heterogeneity within tumor	Hypofractionation marginally beneficial in TC vs. standard treatment	NSCLC and prostate	[56]
System of ODE, TFRT algorithm OAR damage control, no (α,β) heterogeneity, altered daily fractionation	TFRT improves OAR toxicity control vs. standard treatment	Head and Neck	[57]
**Breaking Rule Two**			
System of ODE, evolutionary interplay between OCC and CSC, OAR damage control, intermittent fractionation	Weekly, bi-weekly, or monthly intermittent radiation in one year delays regression vs. standard treatment	glioblastoma	[63]
System of ODE, evolution of emergence of resistance for chemotherapy drugs and radiotherapy, intermittent fractionation	Personalized intermittent hypofractionation improves regression time vs. HFSRT	glioblastoma	[20]
System of ODE, evolutionary interplay between OCC and CSC, concurrent mouse studies, altered daily fractionation	Intermittent hypofractionation prolongs regression in silico and in vivo vs. standard treatment	glioblastoma	[62]
System of ODE, evolutionary interplay between OCC and CSC, clinical applicability	Intermittent hyperfractionation or semi-hypofractionation increases tumor doubling time vs. standard treatment	glioblastoma	[65]

Abbreviations: OS: overall survival, LCC: log cell count, LC: local control, NSCLC: non-small cell lung cancer, TC: tumor control, PDE: Partial differential equation, ODE: Ordinary differential equation, TFRT: temporally fettered radiotherapy, OCC: ordinary cancer cells, CSC: cancer stem cells, HFSRT: hypofractionated stereotactic radiotherapy, OAR: organ at risk.

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
