# Peer review of "A Century of Fractionated Radiotherapy: How Mathematical Oncology Can Break the Rules"

_ijms, 2022, doi:10.3390/ijms23031316_

Round 1

Reviewer 1 Report

Figure â„–1 is missing in the article

Figure â„–3 caption should be revised to be more informative

Too many acronyms. For example, the abbreviation "wild type" WT occurs only twice in the text. “Log Cell Count (LCC)” - the same situation. It is necessary to revise the abbreviations 

Author Response

Thank you for the review. Only the most frequent acronyms have been adapted. Others, such as LCC, have been expanded. The figures have also been updated for accuracy. 

Reviewer 2 Report

The review introduces the concept of mathematical oncology and its application to optimise fractionated radiotherapy. The focus is on the use of mathematical modelling to propose schemes that reduce the effects of radioresistance. Contrary to the title, the review reports nothing really new or revolutionary, known facts are compiled and strung together somewhat differently than usual, but relatively abruptly and only comparatively briefly one finally comes to the topic of insilicio-designed non-normo-fractionated radiotherapy. Here, the clinical data is still very thin (only 19 patients irradiated in this way so far!). Otherwise, the review gives a good overview of the history and background of normo-fractionated radiotherapy and presents current variations of fractionation and dosage based on basic research. The review is well written in itself, and the figures and tables are clear and useful. The relevant papers from basic research are cited, and the bibliography is complete. Reference is also made to the increasing use of hypofractionated and stereotactic irradiation schemes. Here, a reference to the classical stereotactic radiotherapy, the so-called radiosurgery (single dose irradiation), would be desirable. Besides the high ablative dose, the vascular effect on the endothelium seems to have a decisive influence here. In conventional models, radiotherapy controls tumour growth via cellular DNA damage, primarily double-strand breaks, which lead to a reduction in the proliferation potential of the tumour cells and clonogenic survival; in more recent models, the control of tumour growth is predominantly influenced by radiation-induced endothelial cell damage (Garcia-Barros M et al.: Tumor response to radiotherapy regulated by endothelial cell apoptosis. Science 2003; 300: 1155-9.)

Author Response

Thank you for your insight. Our efforts with this review are to drive further investigation in clinical trials utilizing mathematically informed RT fractionation schemas. We agree that SRS is a novel form of therapy that has made its way to clinics that may excite efforts to deliver RT that goes beyond conventional fractionation. A section containing a brief description of stereotactic radiotherapy has been added to clarify its efficacy and we have included in your reference in addition to other articles that describe SRS.